# ViM-UNet: Vision Mamba for Biomedical Segmentation

**Anwai Archit**                                    ANWAI.ARCHIT@UNI-GOETTINGEN.DE
**Constantin Pape**                    CONSTANTIN.PAPE@INFORMATIK.UNI-GOETTINGEN.DE
*Georg-August-University Göttingen, Institute of Computer Science*

**Editors:** Accepted for publication at MIDL 2024

## Abstract

CNNs, most notably the UNet, are the default architecture for biomedical segmentation. Transformer-based approaches, such as UNETR, have been proposed to replace them, benefiting from a global field of view, but suffering from larger runtimes and higher parameter counts. The recent Vision Mamba architecture offers a compelling alternative to transformers, also providing a global field of view, but at higher efficiency. Here, we introduce ViM-UNet, a novel segmentation architecture based on it and compare it to UNet and UNETR for two challenging microscopy instance segmentation tasks. We find that it performs similarly or better than UNet, depending on the task, and outperforms UNETR while being more efficient. Our code is open source and documented at https://github.com/constantinpape/torch-em/blob/main/vimunet.md.

**Keywords:** vision mamba, vim-unet, mamba, ssm, microscopy, instance segmentation

## 1. Introduction

Segmentation is an important task in biomedical image analysis, with applications from radiology to microscopy. Most modern segmentation methods build on CNNs, the UNet (Ronneberger et al., 2015) being most popular. After the success of transformers for text and vision (ViT, (Dosovitskiy et al., 2021)), such architectures have also been proposed for biomedical segmentation; most notably UNETR (Hatamizadeh et al., 2021) and SwinUNETR (Hatamizadeh et al., 2022). They have a global field of view, promising better quality for tasks benefiting from a large context. However, they incur a larger run-time and higher parameter count. More recently, the Mamba architecture (Gu and Dao, 2023), which extends state space models (SSM) (Gu et al., 2022), has been proposed to overcome these computational inefficiencies while keeping a global field of view. It has already been adapted for computer vision by Vision Mamba (ViM) (Zhu et al., 2024).

Here, we introduce *ViM-UNet*, based on ViM, for biomedical segmentation and compare it to UNet and UNETR for microscopy instance segmentation, an important analysis task for biology. Most methods for this task, e.g. CellPose (Stringer et al., 2021), StarDist (Schmidt et al., 2018), are based on the UNet architecture, while recent methods also adopt transformers, e.g. (Archit et al., 2023), and a current benchmark (Ma et al., 2024b) shows favorable results for transformers. We use two different datasets with diverse characteristics, see Fig. 1, and find that ViM-UNet performs comparable or better than UNet (depending on the task) while UNETR underperforms. We validate our results against external methods, nnUNet (Isensee et al., 2021), a well tested UNet framework, and U-Mamba (Ma et al., 2024a), which is also based on Mamba but lacks vision specific optimizations of ViM. Our results show the promise of ViM for biomedical image analysis. We believe that it is especially promising for tasks that rely on a large context, e.g. 3D segmentation or cell tracking.

LIVECell CREMI

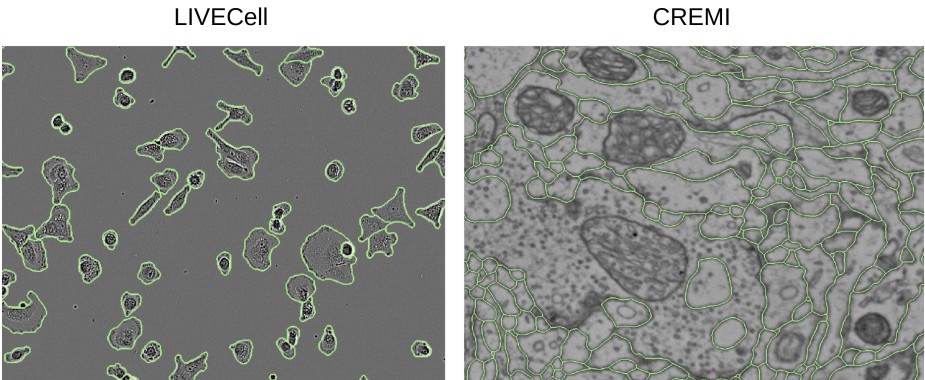

Figure 1: Example images with segmentation from ViM-UNet$_{\text{Small}}$.

## 2. Method and Experiments

We compare three different architectures: UNet, UNETR and ViM-UNet (our contribution), implemented in torch-em (Pape). The UNet has 4 levels and 64 initial features, which double each level. For UNETR we use the ViT of Segment Anything (Kirillov et al., 2023) and the same decoder as UNet, consisting of two conv. layers per level and a transposed conv.; with input from the previous decoder and the ViT output. We chose this simple implementation over skip connections as in UNet and original UNETR, which connect respective levels in encoder and decoder. We found no negative impact of this design. For the ViM-UNet we use a ViM encoder with bidirectional SSM layers. Similar to ViT this model operates on patches; we use a patch shape of 16. The decoder design is identical to UNETR. For UNETR and ViM-UNet we compare encoders of different size; Base, Large, Huge for ViT and Tiny, Small for ViM.

We run comparisons for two datasets: cell segmentation in phase-contrast microscopy (LIVECell (Edlund et al., 2021)) and neurite segmentation in volume electron microscopy (CREMI (Funke et al., 2016)). LIVECell contains small cells with a diverse morphology while CREMI contains neurites of diverse sizes, see also Fig. 1. We restrict the segmentation for CREMI to 2D. For LIVECell we use the given train, validation and test splits, for CREMI we train on the first 75 slices per volume, validate on the next 25 and test on the last 25. For instance segmentation in LIVECell we predict (i) foreground and boundary probabilities, which are post-processed with a watershed and (ii) foreground probabilities as well as cell center and boundary distances, also post-processed with a watershed. We chose (i) to compare with other implementations (see below), where we could not implement distance predictions, and (ii) because this approach is better suited here. For CREMI we use (i). Boundary prediction is standard for this task, usually followed by graph agglomeration (Beier et al., 2017), which is not needed in 2D. The networks are trained for 100k iterations using Adam with initial learning rate of $10^{-4}$ and reduction on plateau. We compare to nn-UNet (Isensee et al., 2021) and U-Mamba (Ma et al., 2024a) with boundary segmentation. Both methods are configured with a hyper-parameter search, for which we use default settings. We use the mean segmentation accuracy (Everingham et al., 2010) for evaluation.

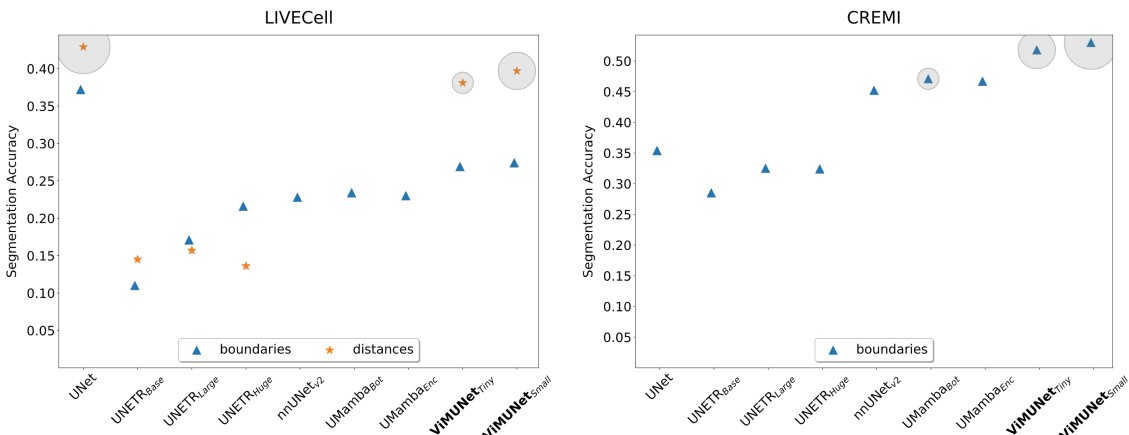

Figure 2: Results for our and external methods; circles highlight the three best methods.

| Methods | UNet | UNETR$_{\text{Base}}$ | UNETR$_{\text{Large}}$ | UNETR$_{\text{Huge}}$ | ViM-UNet$_{\text{Tiny}}$ | ViM-UNet$_{\text{Small}}$ |
|---|---|---|---|---|---|---|
| #Param | 28M | 113M | 334M | 665M | **18M** | **39M** |
| #Memory | $\leq$ 4GB | $\leq$ 24GB | $\leq$ 38GB | $\leq$ 48GB | $\leq$ **9GB** | $\leq$ **10GB** |
| LIVECell | 0.02 (1.2e-4) | 0.15 (3.3e-4) | 0.32 (4.9e-4) | 0.54 (4.6e-4) | **0.05** (2.7e-3) | **0.05** (4.6e-3) |
| CREMI | 0.30 (1.8e-2) | 1.37 (1.8e-2) | 2.95 (3.4e-2) | 4.86 (3.8e-2) | **0.74** (3e-2) | **0.82** (2.4e-2) |

Table 1: Number of parameters, required VRAM for training and inference times per image (in seconds) for our models.

## 3. Results and Discussion

Fig. 2 shows the results. For LIVECell we see that UNet with distance segmentation performs best, followed closely by ViM-UNet. UNETR performs significantly worse. For boundary segmentation UNet clearly performs best, followed by ViM-UNet and the external methods. UNETR again underperforms. For CREMI ViM-UNet performs best, followed by external methods and UNet, with weak results for UNETR. We hypothesize that the global field of view does not bring any advantages for small structures (LIVECell), but that ViM-UNet can leverage it for large structures (CREMI). UNETR underperforms, most likely due to the larger parameter count (see Tab. 1) and lack of pretraining; note that better performance can be achieved through pretraining (Hörst et al., 2023). Comparison to external methods validates that our implementations don't underperform, a fully objective comparison is not possible due to differences in training and inference. We also study inference times and memory requirement for training, see Tab. 1. UNet is the most efficient architecture, followed by ViM-UNet and UNETR.

Overall, ViM-UNet is promising for biomedical image analysis. We believe that it could replace transformer based approaches for applications where large context is important, as it also has a global field of view, but at much higher efficiency. Its lower parameter count enables application on smaller dataset and without extensive pre-training. We plan to extend it to 3D segmentation and tracking, where large context is often crucial.

## Acknowledgments

The work of Anwai Archit was funded by the Deutsche Forschungsgemeinschaft (DFG, German Research Foundation) - PA 4341/2-1. We would like to express our gratitude to Sartorius AG for support in this research through the Quantitative Cell Analytics Initiative (QuCellAI). We also gratefully acknowledge the computing time granted by the Resource Allocation Board and provided on the supercomputer Lise and Emmy at NHR@ZIB and NHR@Göttingen as part of the HLRN infrastructure. The calculations for this research were conducted with computing resources under the project nim00007. We thank Mahdi Enayati for the valuable inputs on vision transformers in microscopy. We thank the authors of ViM (Zhu et al., 2024), Segment Anything (Kirillov et al., 2023), nnUNet (Isensee et al., 2021) and U-Mamba (Ma et al., 2024a) for making their code publicly available.

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
