# OpenReview forum: "ViM-UNet: Vision Mamba for Biomedical Segmentation"
_MIDL.io/2024/Short_Papers — MIDL 2024 Short Papers_

### Official Review · Reviewer_LWiy · 2024-04-24

**Confidence:** 4
**Final Rating:** 5

**Review:**

U-Net and models derived from it have been very popular choices for biomedical image segmentation. With the introduction of transformer based architectures, there has also been transformer based variants of U-Net such as UNETR among others. These models benefit from global views which come at increased computational cost and parameters counts. Therefore, alternatives to transformed  which offer global context but at decreased complex is of interest. Once such model is the Vision Mamba (ViM) architecture. The contribution of this paper is adopting the ViM architecture to build ViM-Unet and test it on two cell segmentation problems. The two cell segmentation problems have very different properties which is a strength in showing the flexibility of the approach. The results are promising in terms of accuracy and computational efficiency in comparison to other models. This is a strong contribution.

---

### Decision · Program_Chairs · 2024-04-26

Accept